# Improved immunoassay sensitivity and specificity using single-molecule colocalization

Amani A. Hariri[1,6], Sharon S. Newman [2,3,6], Steven Tan[4,6], Dan Mamerow[1], Alexandra M. Adams[4], Nicolò Maganzini[3], Brian L. Zhong [4], Michael Eisenstein[1,3], Alexander R. Dunn [4,7] ✉ & H. Tom Soh [1,3,5,7] ✉

Enzyme-linked immunosorbent assays (ELISAs) are a cornerstone of modern molecular detection, but the technique still faces notable challenges. One of the biggest problems is discriminating true signal generated by target molecules versus non-specific background. Here, we developed a **Si**ngle-**M**olecule **C**olocalization **A**ssay (SiMCA) that overcomes this problem by employing total internal reflection fluorescence microscopy to quantify target proteins based on the colocalization of fluorescent signal from orthogonally labeled capture and detection antibodies. By specifically counting colocalized signals, we can eliminate the effects of background produced by non-specific binding of detection antibodies. Using TNF-α, we show that SiMCA achieves a three-fold lower limit of detection compared to conventional single-color assays and exhibits consistent performance for assays performed in complex specimens such as serum and blood. Our results help define the pernicious effects of non-specific background in immunoassays and demonstrate the diagnostic gains that can be achieved by eliminating those effects.

Even after nearly 50 years, the enzyme-linked immunosorbent assay (ELISA) remains an essential tool for the detection of protein biomarkers for both basic research and clinical diagnostics[1,2]. The ELISA has changed relatively little since its inception in 1971—two different antibodies are used to capture and label the target in a "sandwich" format that generates a signal only when both the capture antibody (cAb) and detection antibody (dAb) are bound to the target[1]. This dual-binding requirement confers excellent specificity, but ELISAs remain vulnerable to background signal arising from non-specific binding of antibodies or interferent proteins to the assay substrate[2]. Unwanted background binding can be mitigated to some extent through the use of more stringent wash conditions, blocking of exposed assay substrate surfaces, or careful management of the amount of dAb[3–5]. However, these solutions entail trade-offs in terms of assay performance. For example, overly stringent washing can undermine assay sensitivity by causing loss of signal, while the use of insufficient dAb

concentrations will undermine the assay's ability to accurately resolve target concentrations[6,7].

Consequently, ELISA-based molecular detection is limited by the challenge of discriminating true target-binding events from non-specific background. Researchers have devised a number of different strategies to overcome this difficult problem[8–13]. For example, Chaterjee et al. developed a 'kinetic fingerprinting' assay for the detection of individual surface-immobilized protein molecules[14]. Their approach combines a conventional cAb with a Fab antibody fragment probe for detection[15]. In vitro selection is used to isolate probes that exhibit sufficiently fast dissociation kinetics to achieve rapid, repetitive binding to their target, enabling discrimination of true target recognition events from non-specific background. This technique offers exceptional sensitivity in serum, but only a limited number of probes are currently available and the process of generating novel Fab probes remains challenging and resource-intensive. Zhang et al. reported the

[1]Department of Radiology, Stanford University, Stanford, CA 94305, USA. [2]Department of Bioengineering, Stanford University, Stanford, CA 94305, USA. [3]Department of Electrical Engineering, Stanford University, Stanford, CA 94305, USA. [4]Department of Chemical Engineering, Stanford University, Stanford, CA 94305, USA. [5]Chan Zuckerberg Biohub, San Francisco, CA 94158, USA. [6]These authors contributed equally: Amani A. Hariri, Sharon S. Newman, Steven Tan. [7]These authors jointly supervised this work: Alexander R. Dunn, H. Tom Soh. ✉e-mail: alex.dunn@stanford.edu; tsoh@stanford.edu

use of aptamers for the detection of small-molecule analytes[16,17]. By splitting an aptamer that binds to ATP and labeling each fragment with differently colored fluorophores, they were able to develop an assay that reports binding only when the two fluorophores are in proximity to each other, and thus eliminates background from non-specific binding events. This method is limited by the need for split-aptamer probes that can bind to a target analyte with high affinity and specificity. These reagents remain challenging to engineer, and only a small number of such probes have been described in the literature. Furthermore, this approach has only been applied to small-molecule detection, and it remains unclear how well such an assay would perform with protein targets.

In this work, we describe a two-color sandwich immunoassay that discriminates between specific and non-specific binding and can be applied to a wide range of protein analytes. **Si**ngle-**M**olecule **C**olocalization **A**ssay (SiMCA) employs cAbs and dAbs that have been labeled with distinct fluorophores. The sample is imaged with total internal reflection fluorescence (TIRF) microscopy at sufficiently low concentrations of cAb and dAb such that single molecules of each species can be readily imaged (Fig. 1a). By discarding dAb molecules that are not colocalized with a cAb counterpart, we can greatly decrease the background signal due to non-specific binding, resulting in an improved signal-to-noise ratio, decreased limit of detection (LOD), and increased accuracy for analyte calibration curves. In addition to non-specific dAb binding, heterogeneous cAb surface loading can also contribute substantially to assay variability. Single-molecule imaging allowed us to normalize dAb counts to the counts of cAb for every field of view, thus overcoming this heterogeneity problem. This approach results in far greater sensitivity and consistency of signal across experiments, even in environments with high background. For example, we demonstrated SiMCA with a pair of well-characterized tumor necrosis factor α (TNF-α) antibodies and showed that we could achieve a three-fold lower LOD in serum relative to a non-colocalization-based assay using the same antibodies ($7.6 \pm 1.9$ pM versus $26 \pm 5.8$ pM). Furthermore, our measurements remained consistent whether the assay was performed in buffer, 70% chicken serum, or 70% whole human blood. Collectively, these results demonstrate that SiMCA can overcome the sensitivity and reproducibility limitations imposed by non-specific binding,

enabling accurate detection of picomolar concentrations of protein even in highly complex biological matrices.

## Results

### Overview of SiMCA

SiMCA is a sandwich-based assay in which cAbs and dAbs are each conjugated to a distinct fluorophore tag, and true binding events are indicated only when both fluorescent signals are colocalized (Fig. 1a). As a demonstration, we employed a pair of antibodies that specifically recognize the inflammatory cytokine TNF-α. We labeled the cAb with a green fluorophore (Alexa-546), and site-specifically tagged the antibody with biotin so that it can be immobilized onto a neutravidin-coated surface while ensuring that the antigen-binding domain is appropriately oriented for target binding. The dAb was labeled with a red fluorophore (Alexa-647) (see *Methods*, Supplementary Fig. 1).

For the assay substrate, we passivated a coverslip with a mixture of PEG and PEG-biotin[18] to minimize non-specific binding events and to specifically immobilize the biotinylated cAbs via neutravidin-biotin binding. We then incubated the coverslip with a mixture of TNF-α and the dAb and employed a custom two-color TIRF microscope to acquire images via sequential excitation of the green and red dyes with 532- and 635-nm lasers, respectively. Unbound cAbs are detected solely in the green channel, while dAbs that are non-specifically bound to the substrate are registered only in the red channel (Fig. 1b). In contrast, true binding events give rise to a ternary sandwich complex, resulting in colocalized red and green signals. We used an automated method for image segmentation and registration to count the single-color dAb signals and colocalized binding events in a high-throughput manner across many fields of view (FOVs) per coverslip.

### SiMCA mitigates non-specific binding and improves reproducibility

It is well known that the use of excess concentrations of dAb in an ELISA can contribute greatly to non-specific binding[6,19]. Decreasing the level of dAb employed can mitigate this problem, but at the cost of reduced sensitivity due to loss of signal. SiMCA provides the opportunity to measure the extent of non-specific binding and its impact in terms of background signal, as well as the means to remedy that problem. We first assessed the extent of non-specific binding that

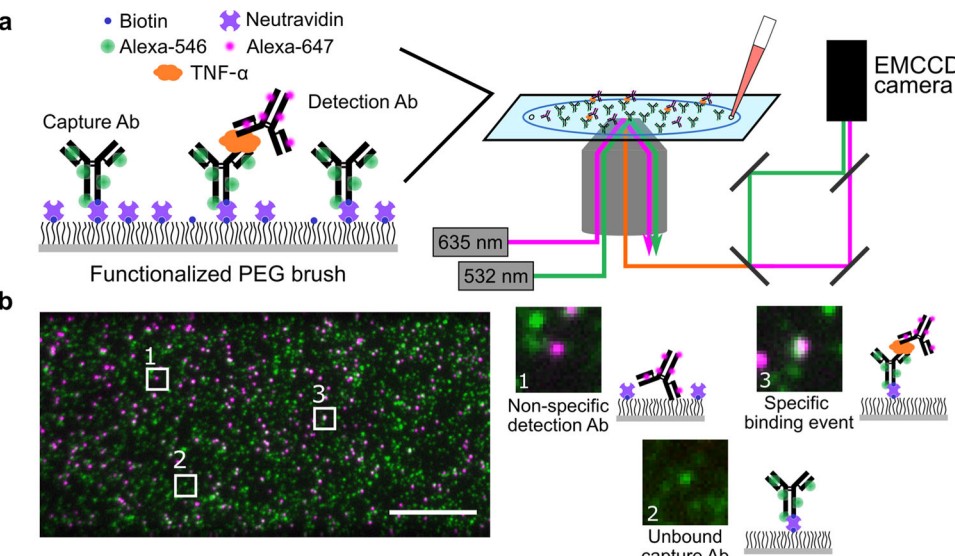

**Fig. 1 | SiMCA platform design. a** A glass coverslip is passivated with a mixture of PEG and PEG-biotin, and then treated with neutravidin and biotinylated, Alexa-546-tagged capture antibodies (cAbs). The surface is then incubated with a solution of the target biomolecule and Alexa-647-labeled detection antibody (dAb). The coverslip is imaged using two-color TIRF microscopy. **b** Images of the cAb and dAb channels are acquired, registered, and analyzed to discriminate non-specific dAb binding (1) and unbound cAbs (2) from true binding events (3). Scale bar = 10 μm.

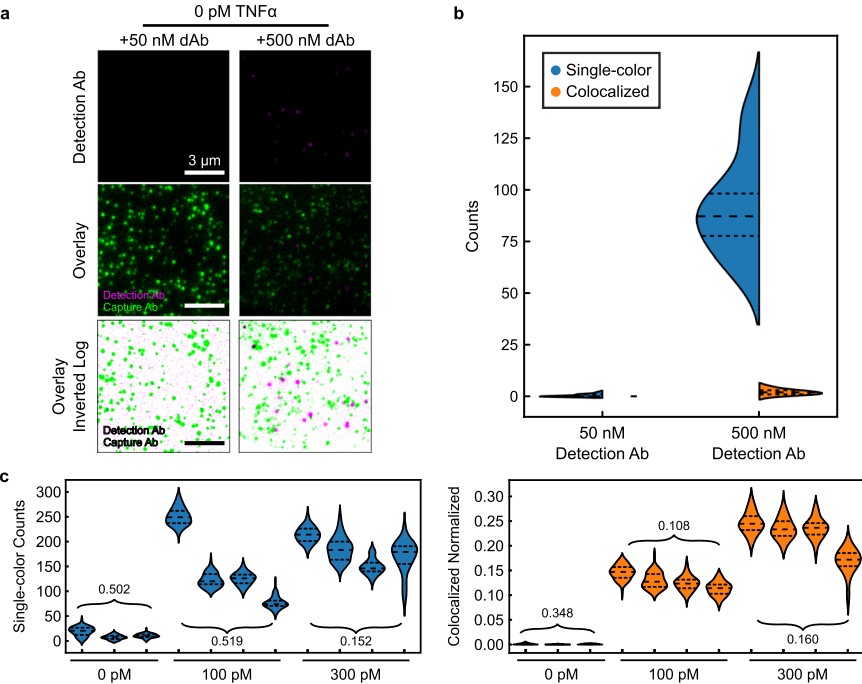

**Fig. 2 | Colocalization minimizes background from excess detection antibody. a** Single-color fluorescence images of dAb only (top), two-color images of cAb and dAb (middle), and log-scale inverted composite images of two-color detection (bottom) in the absence of TNF-α and with 50 nM (left) or 500 nM (right) dAb. Dark spots in the bottom panels represent colocalized signal from the two fluorophores. **b** Distributions of absolute single-color and colocalized counts across 16 fields of view (FOVs). Dashed lines demarcate quartiles of the distribution. **c** Absolute number of dAbs per fields of view (left) and normalized, colocalized counts (right) across different coverslips and TNF-α concentrations. Each violin represents 16 FOVs of a coverslip. Numbers are the coefficients of variance (CV) of the mean FOV counts across coverslips.

occurs at various concentrations of dAb in the absence of TNF-α. We incubated cAb-coated coverslips overnight with either low (50 nM) or high (500 nM) concentrations of dAb, washed the coverslips to remove any unbound dAb, and then acquired 128 (51.2 μm × 25.6 μm) FOVs for each coverslip.

A randomly selected small section of a single FOV shows minimal dAb recruited to the coverslip at 50 nM dAb (Fig. 2a, left). In contrast, at 500 nM dAb, we saw a substantial increase in dAb counts. As there was no TNF-α present, these counts solely represent non-specific binding events. As expected, when we overlaid the two fluorescence channels, most dAb spots did not colocalize with a cAb (Fig. 2a, middle). For visualization, we inverted the color scale, such that overlapping cAb and dAb spots appear as black spots (Fig. 2a, bottom). Any dAb spots that were not spatially overlaid with a cAb are non-specific binding events that would, with conventional methods, be mistakenly counted as a binding event (Supplementary Figs. 2–4). Quantifying the total number of dAbs measured versus the number that colocalized with a cAb across all 128 FOVs of each coverslip revealed that colocalization could eliminate virtually all of these spurious binding events (Fig. 2b; Supplementary Fig. 5). Measured using a single color, mean dAb counts increased from 0.4 ± 0.6 molecules per FOV at 50 nM dAb to 92 ± 23 molecules at 500 nM dAb. In contrast, colocalized dAb and cAb counts were essentially unchanged, increasing only slightly from 0 ± 0.0 molecules at low [dAb] to 2 ± 1.3 molecules at high [dAb]. These results demonstrate that the SiMCA two-color localization strategy can greatly mitigate the effects of non-specific binding.

The vulnerability of single-color methods to non-specific background is further exacerbated by coverslip heterogeneity, wherein the stochastic distribution of cAb on the surface may lead to discrepancies in the number of colocalized pairs observed with identically prepared coverslips and samples. To explore this effect, we incubated multiple coverslips functionalized with cAb with varying levels of TNF-α (0, 100, and 300 pM) and 50 nM dAb. Quantifying the

absolute numbers of dAb counts per FOV across coverslips resulted in a high coefficient of variance (CV) at 100 pM TNF-α (Fig. 2c, left), whereas counting only colocalized events resulted in slightly lower CVs due to reduced background (Supplementary Fig. 6). This modest reduction is attributable to the fact that cAb counts varied considerably both across multiple FOVs within a single coverslip as well as across coverslips (Supplementary Fig. 7). To account for this heterogeneity, we normalized the colocalized dAb counts by the cAb counts in each FOV, where a normalized count of 1 is the theoretical maximum binding and 0 is the theoretical minimum. This normalization greatly decreased the signal variance, with a notable 4.8-fold reduction in CV for the 100 pM coverslips (Fig. 2c, right). At 300 pM TNF-α, the fractional contribution of background dAb signal is expected to be reduced due to the increased number of true binding events. As such, absolute dAb counts and colocalized and normalized counts yielded comparable CVs (Fig. 2c). Since combining colocalization and normalization produces a substantial increase in signal consistency in high-background conditions as well as across coverslips, we have used this analytical approach for the subsequent SiMCA analyses presented below. To prove the general applicability of the SiMCA method, we functionalized a cAb/dAb pair targeting monocyte chemoattractant protein-1 (MCP-1) using the same commercially available fluorophore and biotin kits (See Methods and Supplementary Fig. 8). As with TNF-α, we again demonstrated the capability of SiMCA to eliminate the confounding effects of background produced by non-specific binding of detection antibodies, and greatly improve the reproducibility of our measurements while achieving a pM detection limit.

## SiMCA lowers quantification errors in buffer and serum
Biological specimens such as serum and blood can generate especially high levels of non-specific background due to interferent proteins exhibiting cross-reactivity to dAbs or binding to the assay substrate

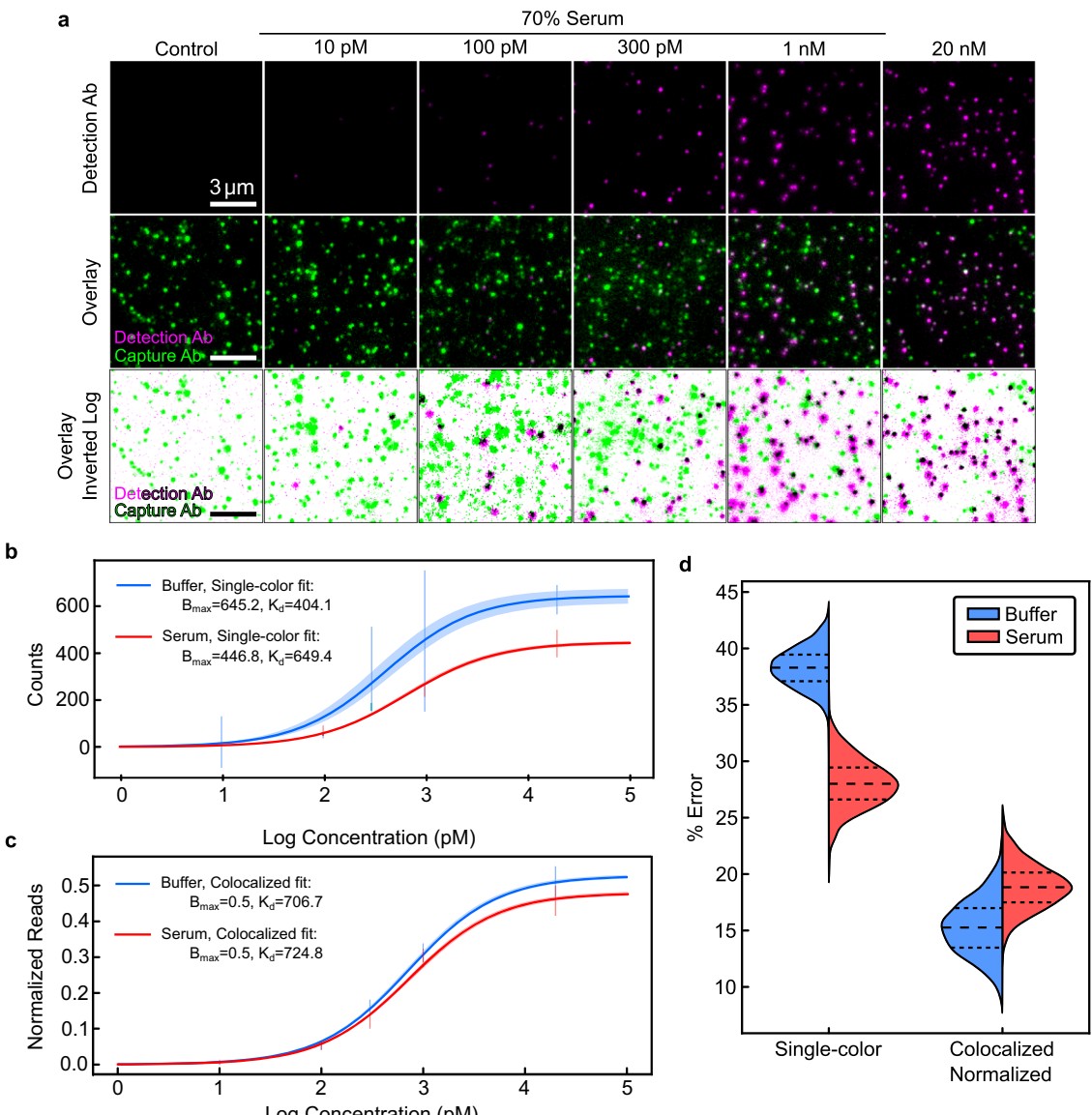

**Fig. 3 | Quantification of TNF-α via single- and two-color analysis in buffer and 70% chicken serum. a** Single-color (top) and colocalized (middle, bottom) signal for increasing concentrations of TNF-α spiked into 70% chicken serum with 50 nM dAb. Best-fit TNF-α binding curves for (**b**) absolute dAb counts and (**c**) normalized, colocalized counts in buffer and 70% chicken serum. 2σ confidence curves are shaded. Summary of primary data represented as vertical bars spanning 1σ from mean. For primary data, please see Supplementary Fig. 9. **d** Bootstrapped mean absolute percent error in TNF-α quantification for buffer and serum for absolute versus normalized, colocalized dAb counts. All distributions are from 128 FOVs collected from two coverslips per condition.

itself. We evaluated how interfering species might impact the quantitative performance and precision of SiMCA versus a conventional single-color assay by comparing single-color and colocalization methods when generating calibration curves for 0–20 nM TNF-α in buffer and 70% chicken serum (Fig. 3a). We used chicken serum to mimic the complexity of human serum without interference from endogenous human TNF-α[20].

In an ELISA, quantification is achieved by fitting a Langmuir isotherm to data from samples spiked with known amounts of target using two fit parameters: an equilibrium dissociation constant ($K_D$) and the maximum specific binding ($B_{max}$). These parameters and their associated uncertainties are used to estimate the concentration of an unknown sample within a certain confidence interval. High confidence is gained when calibration curves have tight parameter fits that are unaffected by the sample matrix. Tighter parameter fits and lower background signal additionally lead to improved ability to resolve lower analyte concentrations, resulting in a lower LOD.

Using absolute dAb counts in buffer, we derived a $K_D$ of $404 ± 35$ pM and a $B_{max}$ of $645 ± 16$ counts (Fig. 3b). In serum, the same analysis yielded markedly different $K_D$ and $B_{max}$ values of $649 ± 17$ pM and $446 ± 3.4$ counts, respectively. The LOD also increased three-fold, from $6.6 ± 1.4$ pM in buffer to $19.4 ± 4$ pM in serum. Surprisingly, the CVs for these parameters were significantly lower in serum than in buffer. Consistent with observations from individual FOVs and coverslips, normalization to cAb counts rectifies this discrepancy, underlining the importance of coverslip-to-coverslip variation as a source of error if left uncorrected. Combining colocalization and normalization yielded narrower confidence envelopes as well as parameter fits for $K_D$ and $B_{max}$ that were virtually identical in serum and buffer (Fig. 3c). In addition, we observed CVs that were 2.8–6.3-fold lower for the fitted parameters relative to values derived from absolute dAb counts, and these were roughly equivalent for serum and buffer experiments (see Supplementary Table 1). We also achieved a lower LOD in serum of $7.6 ± 2.0$ pM, versus $19.4 ± 4$ pM for the single-color measurement.

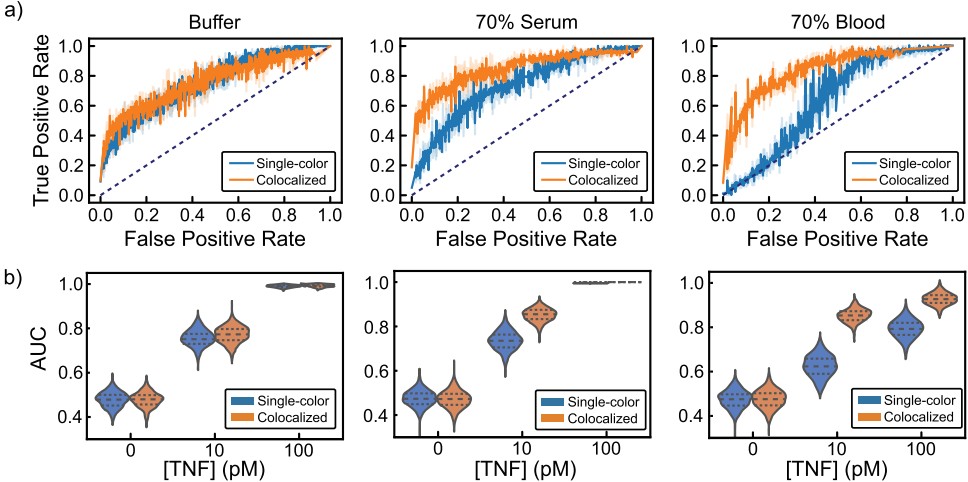

**Fig. 4 | Discriminating TNF-α concentrations using single-color and colocalized methods in buffer, 70% chicken serum, and 70% human blood. a** Bootstrapped ROC curves from binary classification between 10 pM TNF-α distributions and TNF-α-free negative controls in buffer (left), 70% chicken serum (middle), and 70% human blood (right). Dotted lines indicate random guess. Raw data in SI Fig. S9. **b** Area under the ROC curve values from binary classification for 0, 10, and 100 pM TNF-α with 1000 bootstrapped samples. Dashed lines demarcate quartiles of the distribution. All distributions are from 128 FOVs collected from two coverslips per condition.

To estimate errors and the associated confidence in the quantification of unknown TNF-α concentrations, we implemented a bootstrapping approach using the above calibration data in buffer and serum (see Methods). Briefly, we used a subset of the data from each serum and buffer calibration as a training set, with the remaining data used as the test set. Using only the training dataset, we calculated new $K_D$ and $B_{max}$ parameter fits. We then used these fit parameters to predict sample concentrations for the test data set. The mean error was then calculated from the predicted versus true concentrations for the test set. We then repeated the process of splitting, fitting, testing, and calculating errors 1000 times to obtain confidence intervals in the errors (see Methods). As shown in Fig. 3d, normalized and colocalized dAb data reduced errors relative to single-color dAb counts by 2.5-fold and 1.5-fold for buffer and serum respectively (see Supplementary Table 2). Notably, the errors with colocalization remained similar in buffer and serum.

## SiMCA lowers false-positive rates in complex samples
Finally, we set out to quantitively evaluate the diagnostic sensitivity and specificity of the single-color versus the two-color colocalization approach in a series of assays performed with low concentrations of TNF-α (0, 10, and 100 pM) spiked into either 70% chicken serum or 70% human blood. For reference, physiological concentrations of TNF-α range from 4 pM at baseline[21] to a mean of 40 pM and up to 300 pM for patients developing septic shock[22]. In our experiment, absolute dAb counts varied widely in both serum and blood, with some distributions heavily skewed with long tails or even bimodal—this reflects the inherent heterogeneity of the coverslips, as discussed above. In contrast, normalized, colocalized counts were markedly more consistent across buffer, serum, and blood (see Supplementary Fig. 10).

To evaluate the usefulness of colocalization and normalization in the context of a potential clinical application (i.e., detection of TNF-α), we conducted separate binary classifications between distributions with TNF-α and the control distribution without TNF-α. This enabled us to characterize the trade-off between true positive rates (TPR, or sensitivity) and false-positive rates (FPR, or 1−specificity) via receiver operating characteristic (ROC) curves. For reference, an ideal assay with little to no overlap between the target and control distributions would achieve high TPR with a low FPR−approaching perfect discrimination in the upper left corner (TPR = 1, FPR = 0). An assay with poor discrimination (i.e., producing similar distributions for both classes) would have an ROC curve closer to the diagonal (dotted line in Fig. 4a), equivalent to random guessing.

At 10 pM TNF-α, the ROC curves for the absolute dAb counts and colocalized, normalized methods were similar for samples prepared in buffer (Fig. 4a). However, the colocalized, normalized data were clearly superior to those from absolute dAb counts in 70% serum or whole blood, particularly at lower FPRs, further highlighting the inability of single-color methods to distinguish false positives resulting from background dAb binding and coverslip variability (Fig. 4a; Supplementary Fig. 10). In addition, the colocalized, normalized ROC curves were largely identical across buffer, serum, and whole blood (Fig. 4a). To quantitatively compare the ROC curves across sample matrices and TNF-α concentrations, we calculated the area under the curve (AUC) (Fig. 4b). Increased TNF-α concentrations yielded an appreciable increase in AUC, as would be expected when distributions are being pushed further apart by increased numbers of true binding events. However, AUC values calculated from the single-color dAb counts were significantly lower than those derived from colocalized, normalized counts in both serum and blood. This difference is particularly apparent at 10 pM TNF-α, where non-specific recruitment of dAb to the coverslip surface accounts for a considerable fraction of total counts. In summary, SiMCA achieved consistent and accurate analyte detection that was robust against background signal arising from complex biological specimens compared to a conventional single-color approach.

## Discussion
In this work, we demonstrate that SiMCA can overcome the confounding effects of non-specific background to enable accurate, sensitive, and reproducible detection of picomolar protein concentrations even in highly complex sample matrices. We evaluated SiMCA using well-characterized, commercially available TNF-α antibodies, and found that the use of colocalization and normalization reduced variability and achieved a more consistent signal across coverslips, yielding CVs that were 2.8–6.3-fold lower than those derived based on absolute, single-color dAb counts. Our approach also produced quantification values that remained consistent in different sample matrices, with narrower confidence envelopes, more consistent parameter fits between serum and buffer, and a consistently lower LOD even in complex samples such as chicken serum and human

blood. Thus, our technique provides a generalizable way to achieve more robust immunoassay performance.

As with any assay, SiMCA does suffer from some limitations. At present, SiMCA requires relatively expensive microscopy equipment that can achieve single-molecule sensitivity. Extending the benefits of SiMCA to lower-resource environments would require strategies to boost the fluorescence signal to levels that can be detected by smartphone cameras[23]—for example, by using fluorescent nanoparticles that emit a substantially brighter signal, or fluorescence-enhancing materials[24] that maximize the output from individual fluorophores.

Our focus in this work was to understand and reduce general sources of error in immunoassays, rather than to demonstrate sensitivity that outperforms existing molecular detection assays. We note that in theory the sensitivity of SiMCA is limited primarily by the number of dAbs and cAbs counted, as well as the accuracy with which colocalization is determined. The former quantity can be addressed simply by scanning larger FOVs on the coverslip; in the present study, we examined only 0.5% of the flow cell surface. In our current set-up, the EMCCD camera's field of view was adjusted to split the channels to show parallel images for single-molecule Förster resonance energy transfer (FRET) experiments (described below). Scanning larger areas is also possible but may impact the assay's performance by increasing imaging time, such that the effects of antibody dissociation will become more meaningful. Future studies might explore a post-crosslinking approach, which would allow coverslips to be scanned for tens of minutes and thereby achieve better sensitivity. Increasing colocalization accuracy helps to eliminate false positives, while also allowing higher cAb densities (Supplementary Fig. 11). In its current form (Supplementary Fig. 11, *left*), the assay's primary limitation on sensitivity is its low [cAb] (~2 pM), resulting in low capture efficiency of target molecules[25]. This low [cAb] is required in SiMCA's current format due to the coating density limitations imposed by diffraction-based identification and localization of cAbs. Future approaches that enable higher effective [cAb] levels (Supplementary Fig. 11, *right*), and thus higher target capture efficiencies, could be achieved by strategies including increased surface area-to-chamber volume ratios and super-resolution imaging techniques that achieve sub-diffraction-limited molecular localization (e.g., STORM/STED[26], DNA-PAINT[27], FRET[28], etc.).

As single-molecule fluorescence colocalization can be determined with Ångstrom precision, this suggests that the ultimate limit on colocalization in our assay is the size of the antibodies used (~10 nm). We note as well that FRET provides an alternate, stringent test of fluorophore colocalization. Indeed, we observed FRET between colocalized dAbs and cAbs (Supplementary Fig. 12), suggesting a promising means of increasing the sensitivity and specificity of SiMCA. Finally, we would note that SiMCA, like other immunoassays approaching single-molecule detection, is limited by molecular shot noise, where the theoretical sensitivity is statistically dictated by unavoidable Poisson error[29].

Most antibodies with sub-micromolar $K_D$ values should be compatible with SiMCA. However, with lower-affinity antibodies in the micromolar-millimolar range, there is a legitimate concern that the low cAb surface coverage might impede SiMCA performance. This could be an issue for detecting certain ligands, such as some small molecules and peptides. In order to generalize our assay to low-affinity antibodies, future studies may examine potential solutions including: (1) increasing cAb density, which will result in higher capture efficiency (avidity[30]), (2) decreasing imaging time, and (3) improving assay kinetics through the use of crowding agents or cross-linking of antibody-target pairs.

Despite these limitations, SiMCA has multiple advantages in comparison to other single-molecule imaging-based methods such as SiMoAs, iSCAT, iSCAMS and other state-of-the-art techniques like SPR

—most notably, its robustness against non-specific binding[19,31–33]. SiMoAs immunoassays capture microscopic beads decorated with specific antibodies and then label the immunocomplexes with an enzymatic substrate that generates a fluorescent product. Fluorescence imaging is then used to image single beads, each confined in femtoliter-scale reactions chambers, for the purpose of detecting single protein molecules. This approach can detect as few as ~10–20 enzyme-labeled complexes in a 100 μl sample and allows detection of clinically relevant proteins in serum at femtomolar concentrations and less, which is much lower than conventional ELISA. However, unlike SiMCA, this assay does not provide the ability to isolate and interrogate single molecules on individual beads, and thereby distinguish true antibody-antigen binding events from non-specifically bound complexes.

Methods such as iSCAT, iSCAMS and SPR offer the powerful advantage of label-free imaging. iSCAT and iSCAMS can both deliver real-time imaging of single unlabeled biomolecules. In solution, biomolecules scatter and reflect light upon continuous illumination with coherent light. The contrast resulting from the scattered and reflected light interference at the detector is leveraged in both iSCAT and iSCAMS to enable single-particle imaging. While powerful, both techniques are potentially prone to spurious detection events due to nonspecifically adsorbed macromolecules, a consideration that may be particularly important in complex media. SPR has been widely used for the measurement of biomolecular interaction kinetics in real-time. SPR detects changes in the reflected light when an analyte binds to (or unbinds from) the sensor surface, making this technique label-free and direct. Immobilization of the analyte-binding substrate in SPR is achieved by adsorption onto gold surfaces, in contrast to SiMCA, which uses PEG passivation and biotin-streptavidin to specifically immobilize cAbs. Although the techniques differ in their immobilization strategies, SPR and SiMCA offer similar sensitivities for detection. However, because SPR is a label-free method, it cannot confidently distinguish between specific binding of the analyte versus other biomolecules from serum, blood, or other complex media. In contrast, SiMCA is well suited for detecting analytes directly from serum and blood due to the inherent specificity provided by two-color colocalization.

## Methods

### Materials and buffers

mAb1 (cAb) and mAb11 (dAb) anti-TNF-α antibodies were purchased from Biolegend. Human TNF-α protein was purchased from R&D Systems (210-TA). Human blood was purchased from BioIVT. Unlabeled mouse anti-human MCP-1 antibody (clone 5D3-F7; BDB551226), unlabeled mouse anti-human MCP-1 antibody (clone 10F7; BDB555055), and recombinant human MCP-1 (BDB554620) were purchased from BD Biosciences. The SiteClick biotin antibody labeling kit, Alexa Fluor 546 NHS ester (succinimidyl ester) and Alexa 647 NHS ester (succinimidyl; ester), and all other chemicals were purchased from Thermo Fisher Scientific. All chemicals were of analytical grade and used without further purification. 1% v/v Vectabond was purchased from Vector Laboratories. PEG succinimidyl valerate MW-5000 (mPEG-SVA) and biotin-PEG-SVA MW-5000 (Biotin-PEG-SVA) were purchased from Laysan Bio. Custom imaging chamber components were purchased from Grace Bio-Labs. Coverslips were purchased from Thermo Fisher Scientific. The PBST buffer (pH 7.4) used in these experiments contained 137 mM NaCl, 2.7 mM KCl, 10 mM $Na_2HPO_4$, 1.44 mM $KH_2PO_4$, and 0.1% Tween 20. The 1X PBSBT buffer contained 137 mM NaCl, 2.7 mM KCl, 10 mM $Na_2HPO_4$, 1.44 mM $KH_2PO_4$, 0.1% Tween 20, and 1% BSA.

### Sample preparation and imaging

Coverslips were soaked in piranha solution (25% $H_2O_2$ and 75% concentrated $H_2SO_4$) and sonicated for 1 h, followed by multiple rinses in

water (Thermo Fisher Scientific, molecular-biology grade) and acetone (Thermo Fisher Scientific, HPLC grade). Dry and clean coverslips were then treated with Vectabond/acetone (1% v/v) (Vector Labs) solution for 5 min and then rinsed with water and left in a dried state until used. In order to prevent non-specific adsorption of biomolecules onto the glass surface, coverslips were functionalized prior to use with a mixture of poly(ethylene glycol) succinimidyl valerate, MW 5000 (mPEG-SVA) and biotin-PEG-SVA at a ratio of 99:1 (w/w) (Laysan Bio) in 0.1 M sodium bicarbonate (Thermo Fisher Scientific) for 3 h[18]. Excess PEG was rinsed with water, and the coverslips were dried under a $N_2$ stream. Imaging chambers (~5 µL) were constructed by pressing a polycarbonate film (Grace Bio-Labs) with an adhesive gasket onto a PEG-coated coverslip. Two silicone connectors glued onto the pre-drilled holes of the film served as inlet and outlet ports. The surface was incubated with 7 µL of a 2 mg/ml neutravidin solution (Thermo Fisher Scientific). Excess neutravidin was then washed off with 100 µL of 1X PBS buffer.

## TIRF microscopy
Single-molecule fluorescence measurements were performed with objective-type TIRF microscopy on an inverted microscope (Nikon TiE) with an Apo TIRF 100× oil objective lens, NA 1.49 (Nikon) as described previously[34], and controlled using Micro-Manager[35]. Samples were excited with a 532 nm (Crystalaser) or 635 nm (Blue Sky Research) laser. Excitation light was cleaned with a quad-edge laser-flat dichroic with center/bandwidths of 405/60 nm, 488/100 nm, 532/100 nm, and 635/100 nm from Semrock (Di01-R405/488/532/635-25×36), and the emission signal was passed through the corresponding quad-pass filter with center/bandwidths of 446/37 nm, 510/20 nm, 581/70 nm, 703/88 nm (FF01-446/510/581/703-25). The emission signal was then separated using a dichroic beam splitter (635 nm), passed through an additional set of filters (546 channel: 593 nm/40 nm (Semrock); 647 channel: 675/30 nm (Semrock), and recorded on an EMCCD camera (Andor iXon), as described previously[34]. We captured 16-bit 512 × 512 pixel images with an exposure time of 200 ms, and a multiplication gain of 2800–3000. Excitation was carried out at a full power setting (25 mW) with a power output of 2–3.5 mW at the objective for the green (532 nm) laser. The excitation power of the red laser ranged between 2 and 4 mW at the objective based on the experiment. We typically observed 150–300 spots per 35 µm × 70 µm field of view.

We used a custom-made polycarbonate chamber with dimensions of 13 mm × 4 mm × 150 µm, which matches the size of the cAb-functionalized area. The concentration of cAbs on the surface was estimated to be around 2 pM. We collected assay data by rastering a 400 µm × 400 µm area of the pegylated region of the coverslip at 5 µm intervals, producing 64 images per coverslip per channel. Two sets of 64 images were collected for each TNF-α concentration. The collection of these images, which each included green (200 ms) and red (200 ms) channels (200 ms) plus a blank (for oil equilibration, 2.5 s), took ~3 min in total, and provided sufficient precision for all samples tested over the 0.01–10 nM TNF-α concentration range studied (Supplementary Fig. 13) shows the calculated CVs of dAb and cAb counts after sampling 3–100 FOVS for eight coverslips (bootstrapped). We can see that a plateau is reached well before the 64 FOV mark.

We limited the number of FOVs (64 FOVs, 0.5% of the flow cell surface) scanned to minimize imaging time (~3 min)—and thereby minimize the effects of dissociation on the sensitivity of our assay. We are aware that scanning larger FOVs on the coverslip would improve assay sensitivity. One solution would be to switch to a set-up with a larger FOV; this would be challenging with our current set-up, in which the EMCCD camera FOV has been adjusted to split the channels to show parallel images for single-molecule FRET experiments. Scanning larger areas is also possible, but may impact assay performance by increasing imaging time to an extent that dissociation/off-rate becomes a meaningful factor.

Using SPR, flow cytometry, and previously reported data on the $K_d$ of the antibodies[36,37](mAb1 and mAb11) utilized in this study, the $K_d$ values for mAb11 and mAb1 are in the range of ~1.0–2.0 nM (Supplementary Fig. 14) and the reported average $k_{on}$ for an antibody is ~$10^5 M^{-1}s^{-1}$. With such high affinities, we expect minimal dissociation ($K_d = k_{off}/k_{on}$) within the reported imaging time (~3 min).

## Antibodies
The anti-human TNF-α cAb (mAb1) was functionalized in house. The antibody was first biotinylated site-specifically using the SiteClick biotin antibody labelling kit according to the manufacturer's instructions. The SiteClick biotin antibody labeling kit specifically attached the biotin to the heavy chain of the antibody, targeting the carbohydrate domains present on essentially all IgG antibodies and thereby ensuring that the antigen-binding domains remain available for binding to the antigen target. The antibody was then labeled with the Alexa Fluor 546 antibody labeling kit (Thermo Fisher Scientific, A20183) according to the manufacturer's instructions (Supplementary Fig. 1). Anti-human TNF-α dAbs (mAb11) were pre-labelled with Alexa Fluor 647 (67 nM stock solution, pre-concentrated at 800 nM for high concentration experiments) and used at the indicated concentrations. The same labelling protocols were applied to the MCP-1 mouse anti-human antibodies (clone 5D3-F7 was labeled with Alexa 546 fluorophore and site-specific biotin at 0.31 mg/ml, and clone 10F7 with Alexa 647 fluorophore at 0.63 mg/ml). Quality control experiments confirmed that binding affinity of antibodies was not compromised by labelling. Furthermore, we determined that the degree of labeling (DOL) for all antibodies was three to five dye molecules per antibody, with an estimated Poisson distribution shown in Supplementary Table 3. This was calculated according to the following equation: $[P(x, DOL) = (e^{-DOL})(DOL)^x/x!]$. Importantly, the probability of having no dye molecules coupled to a given antibody is close to zero, and the detection of even single dye labels is well-established using single-molecule fluorescence. Non-specific background was caused mainly by dAbs and cAbs adhering to the coverslip surface, with minimal surface binding by TNF-α even at very high target concentrations.

## Choice of fluorescent labels
We chose Alexa 546 for this assay because it has greatly improved photostability over Alexa 532 under our experimental conditions, where rapid photobleaching and/or blinking could compromise data analysis and event counting. We found that the average survival time for Alexa 546 was notably larger than that of Alexa 532—14 s versus 3 s—in the assay conditions that we chose to minimize blinking. We thus concluded that Alexa 546-labeled cAbs will not photobleach within a time-frame that would interfere with the counting process (Supplementary Fig. 15).

## Detection of TNF-α
Neutravidin-coated coverslips were first incubated with 7 µL of 0.3 nM biotinylated Alexa 546-labeled cAb solution for 5 min, then washed using 1X PBST buffer to get rid of unbound material (i.e., concentration of Alexa 546-labeled cAb stock was 2.3 µM). Images of coverslips before and after cAb addition were acquired to account for background noise, channel leakage, and donor fluorophore intensity (see Supplementary Fig. 12). Subsequently, we added 7.5 µL of 50 nM dAb solution spiked with recombinant human TNF-α at different concentrations (0, 0.01, 0.1, 0.3, 1, and 20 nM). The TNF-α was prepared as 25 µl samples in 1.5 mL Protein LoBind tubes (Eppendorf). TNF-α was thawed on ice and added at 5.9× (for 0.01, 0.1, 0.3, and 1 nM) or 2.95× (for 20 nM) the final concentration. PBSBT was added to a final volume 17.5 µl, after which 7.5 µl of dAb was added for a final dAb concentration of 50 nM. The samples were immediately injected into the prepared coverslips and incubated covered overnight at 4 °C. For chicken serum and human blood experiments, we replaced the buffer in the above

protocol with chicken serum or human blood, where the final serum and blood percentage after mixing with dAb was 70%. The fluorescent background greatly increased in the presence of serum or blood in the coverslip due to autofluorescence/quenching effects, but an additional washing step restored the initial signal/background ratio. Supplementary Fig. 16 shows that the distribution of cAb intensities and counts remained constant following overnight incubation with buffer and serum. This demonstrates the robustness of the surface passivation layer, avidin linkage, and fluorophore-antibody complex.

We note that because TNFα is a small protein, we could detect FRET between the donor fluorophore on the cAb and acceptor fluorophore on the dAb upon binding the protein target. Because we were working with Alexa 546 and Alexa 647, with an Förster radius $R_0 = 8$ nm, we were able to detect energy transfer and compute FRET efficiencies upon protein binding. This was confirmed by the drop in donor intensity and increase in acceptor intensity upon target binding (Supplementary Fig. 12).

## Image segmentation and registration

We stitched together an image to directly map cAbs and dAbs from the green- and red-only excitation images (green left, red right). We denoised by performing a Gaussian filter using a standard deviation for the kernel of 0.8. We then isolated regional maxima. Since the raw images can be unevenly illuminated, we first subtracted from the Gaussian-filtered image a background image obtained by morphological reconstruction, as described in the *scikit-image* example[38]. The resulting background subtracted image was then mapped to a more dynamic intensity range by an inverse hyperbolic sine transformation. Local maxima and their respective x, y coordinates were selected if their intensity values were at least 1.2 times the standard deviation plus the median. cAb and dAb spots were differentiated by locations of the detected local maxima in the left and right halves of the stitched image, respectively. To determine the colocalized spots, we mapped the coordinates of the dAb spots to the region of the cAb using a pre-defined affine transformation (see below). The transformed dAb coordinates (dAb') were then matched to the true detected cAb coordinates. If a pair of coordinates (cAb[i], dAb'[j]) were within a Euclidean distance of 1.5 pixels, the pair was counted as a colocalized spot.

The affine transformation matrix was made per experiment day to account for any misalignment of the microscope setup. First, 100 nm Tetraspeck beads (Thermo Fisher Scientific T7279) were rastered across a glass coverslip. Five to ten images were taken, and a transformation matrix was determined for each image and averaged to produce the final transformation matrix. Capture and detection images were stitched and localized as described above. Capture and detection coordinates were matched up to their closest neighbors across all coordinates using a KD tree with leaf size of 30 and using the Euclidean distance metric. Coordinates in both images were saved as C and D matrices, respectively, of size $n \times 3$ where $n$ is the number of beads detected. The coordinates $c_0, ..., c_n$ were projected onto the corresponding coordinates $d_0, ..., d_n$ using the affine transformation matrix T:

$$\begin{bmatrix} D \\ 1 \end{bmatrix} = T \begin{bmatrix} C \\ 1 \end{bmatrix} \quad (1)$$

where

$$T = \begin{bmatrix} d_0 \cdots d_n \\ 1 \end{bmatrix} \begin{bmatrix} c_0 \cdots c_n \\ 1 \end{bmatrix}^{-1}$$

The average of T matrices was calculated for each of the bead images and provided for transformation of experimental data.

Finally, we applied a quality filtering step. If the spots were not well-distributed, this could reflect faulty frames of view due to bubbles, large dust particles, etc. Since the dynamics of target and dAb binding to the coverslip surface could be significantly altered in these frames, we removed these in this post-processing step. The mean y coordinates of the spots detected in the left half of the image were averaged. If the averaged coordinate was not within 75 pixels of the middle of the image, that frame of view was removed from analysis. Less than 1% of frames of view were discarded because of this filter.

By implementing tighter colocalization criteria of 10–40 nm, either statistically and/or with advanced imaging, we would expect drastically improved sensitivity and a decrease in the number of false-positive events. This is also influenced by the amount of cAb on the coverslip surface. Higher cAb density improves assay sensitivity by providing more binding sites, but will also confound the discrimination of fluorescent spots and introduce errors in single-molecule counting in a diffraction limited set-up. To determine the balance of optimal spatial resolution and capture density needed to maximize the sensitivity of our assay, we performed simulations for different colocalization cutoff distances (10–300 nm) to estimate the number of false colocalization events as the number of non-specific binding events increases (Supplementary Fig. 17). Shorter distances would reduce the number of false colocalizations, but also requires higher spatial resolution. For example, with our current colocalization criteria (~200 nm), we would expect ~4.5 of every 100 non-specific binding events to be counted as a binding event. With a distance of 100 nm, we would estimate this number to be ~1− theoretically reaching 0 at a distance of 10 nm. We also confirmed that increasing the number of cAbs on the surface would increase the number of false colocalizations, but could also lower the assay LOD. The smallest simulated cutoff distance (10 nm) could be achieved through FRET-based detection or with super-resolution techniques and more sophisticated software and imaging techniques.

## Absolute single-color and normalized, colocalized analysis

Absolute single-color dAb spots were simply localized and counted in the Alexa-647 channel. However, this counting is highly dependent on the consistency of cAb coverage across coverslips and frames of view. For SiMCA colocalized spot analysis, we addressed coverslip variability by normalizing spot counts relative to the cAb signal in the Alexa-546 channel:

*Normalized colocalized counts* = [*absolute colocalized counts*]/ [*cAb counts*].

For comparison of just normalization methods to absolute single-color dAb spots in our SI, we also normalized single-color counts as such:

*Normalized single-color counts* = [*absolute dAb counts*]/[*cAb counts*].

We found normalization lowers quantification error, and thereby enable us to exploit colocalization to reduce the effects of non-specific binding.

## Fitting to binding curve

To create a calibration curve, we fitted our data to the Langmuir binding isotherm:

$$Counts = \frac{B_{max} * [TNF\alpha]}{K_d + [TNF\alpha]} \quad (2)$$

where counts can be single-color unnormalized counts, normalized two-color counts, or normalized, colocalized counts, and $[TNF\alpha]$ is the concentrations used in the binding curves (10 pM, 100 pM, 300 pM, 1 nM, and 20 nM). $K_d$ (equilibrium dissociation constant) and $B_{max}$ (maximum signal possible) were determined by the curve-fitting function in python using Scipy's 'optimize curve fit' function, which

uses non-linear least squares to fit a function. To calculate the concentration given a signal, we then used the inverse function with fit parameters $K_d$ and $B_{max}$ as follows:

$$[TNF\alpha] = \frac{counts - K_d}{B_{max} - counts} \tag{3}$$

## Calculation of LOD

As per convention, we defined LOD as the signal that is three standard deviations ($\sigma_y$) above the mean signal ($\bar{y}$) obtained without analyte: $LOD = \bar{y} + 3*\sigma_y$. We then converted this value to its associated concentration using the fits from the binding curve and the inverse binding curve function (Eq. 3). The corresponding error in LOD is determined by propagation of errors of the inverse function ($f$) and errors associated with the binding curve fits:

$$\sigma_{LOD} = \sqrt{\left(\frac{df}{dB_{max}}\sigma_{Bmax}\right)^2 + \left(\frac{df}{dK_d}\sigma_{Kd}\right)^2 + \left(\frac{df}{dy}\sigma_y\right)^2}$$
$$= \sqrt{\left(\frac{K_d*y}{(B_{max}-y)^2}\sigma_{Bmax}\right)^2 + \left(\frac{y}{B_{max}-y}\sigma_{Kd}\right)^2 + \left(\frac{K_d*B_{max}}{(B_{max}-y)^2}\sigma_y\right)^2} \tag{4}$$

## Quantification with bootstrapping

We bootstrapped our data 1000 times to properly characterize mean errors and confidence in quantification of new unknown targets. For each bootstrap iteration, we sampled with replacement from our dataset of 128 FOVs, fit to the binding curve in Eq. 2, then used the remaining unsampled data to predict the TNF-α concentration for each FOV (Eq. 3). We calculated the error between the predicted and true concentration of these test samples using Mean-Absolute-Percentage-Log-Error (MAPLE):

$$MAPLE = \frac{100\%}{\#\,test\,samples}\sum \left|\frac{\log(True[TNF]) - \log(predicted[TNF])}{\log(True[TNF])}\right| \tag{5}$$

Finally, after all bootstrap iterations, we calculated the standard deviation of MAPLE.

## Bootstrapped binary classification, ROC, and AUC calculations

We conducted binary classification given two distributions to quantitatively evaluate diagnostic sensitivity and specificity. First, we combined the two distributions into one dataset with N frames of view: $\{(x_1, y_1),...,(x_N, y_N)\}$ (typically N = 128). A given single-color or colocalized count, $x_i$, is assigned to a label, $y_i$, where $y_i$ is 0 if it is the 0 pM control or 1 if it is a sample with TNF-α. We then split the dataset into training and test datasets, $T_{train}$ and $T_{test}$. Using the sklearn python library[39], we fitted our training data to a binary logistic regression classifier without regularization, such that we minimize the following cost function with respect to the parameters $w$ and $c$:

$$Loss = \sum_{i=1}^{n} \log\left(\exp\left(-y_i*(x_i*w+c)+1\right)\right) \tag{6}$$

Using the $T_{test}$ dataset and sklearn libraries, we then calculated the probability estimates and respective ROC curve and AUC values. To estimate confidence in the ROC and AUC values, we bootstrapped the above binary classification 1000 times. For each bootstrap iteration, we sampled each distribution with replacement for the training set. The remaining unsampled data were then used as the test set. The above classification and ROC/AUC metrics were then calculated.

## Statistics and reproducibility

We determined that 16 frame of views (FOVs) provided minimum sufficiency for each channel/coverslip by analyzing bootstrapped variance of a range of FOVs from three coverslips from our initial experiment. We calculated that variance of the mean spot counts across FOVs dropped significantly beyond using 16 FOVs, and consistently plateaued before 64 FOVs. We chose 64 as a safe cut off much beyond the inflection point as shown in Supplementary Fig. 13.

Coverslips were excluded from analysis when impurities/contamination were present in the sample, which was indicated by the presence of micron size bright fluorescent spots covering large areas of coverslips surface. This could be due to buffer or Ab sample contamination. FOVs were removed due to bubbles or large dust particles being introduced at the FOV. Since the dynamics of target and dAb binding to the coverslip surface could be significantly altered in these frames, we removed these in an automated post-imaging step described in the Image segmentation and registration section in the methods.

**Replication.** A minimum of two coverslips were prepared, imaged and analyzed for each condition. A total of 16 or 64, FOVs were acquired and analyzed for each coverslip. More replicates were performed for the study, but due to laser damage/change and microscope/camera re-alignment issues, they were dismissed from the study for consistency. All duplicates presented in this work were done using the same laser/power, exposure time / Gain and same microscope/camera alignment. All attempt at replications were successful.

**Randomization.** Coverslips were randomly allocated for TNF-alpha/dAb and MCP-1/dAb samples of different concentrations.

**Data collection.** Blinded. Samples were prepared in the Soh lab (numbered, and each number was associated to a concentration written in notebook in Soh lab) and imaged afterwards in Dunn lab. Concentrations were revealed after imaging.

**Data analysis.** Blinded. FOVs per coverslip were analyzed automatically using custom image analysis software without bias from investigator. Only after spot counting was done, were concentrations revealed.

## Reporting summary

Further information on research design is available in the Nature Research Reporting Summary linked to this article.

## Data availability

The data generated in this work are available in the Stanford Digital Repository (https://purl.stanford.edu/bc494tq1762). A subset of data for demonstration purposes are available at https://github.com/newmanst/simca-pub. The source data underlying Figs. 2b, d, 3b, c, d, 4a and Supplementary Figs 5, 6, 7, 8, 9, 10, 13, 14, 15b, c, and 16 are provided as a Source Data file. Source data are provided with this paper.

## Code availability

Code is available here: https://github.com/newmanst/simca-pub

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

## Acknowledgements

This work was supported by the Chan-Zuckerberg Biohub (H.T.S.), the Helmsley Trust (H.T.S.), and the National Institutes of Health (NIH, OT2OD025342 (H.T.S.), R01GM129314-01 (H.T.S.), and R35GM130332 (A.R.D.)). A.A.H. acknowledges support from the Sanjiv Sam Gambhir—Philips Fellowship Program in Precision Health and the NSERC Postdoctoral Fellowships (PDF, Canada). S.S.N. acknowledges support from SGF (Stanford Graduate Fellowship in Science & Engineering) and the NSF Graduate Research Fellowship Program (GRFP).

## Author contributions

H.T.S., A.A.H., and A.R.D conceived the initial concept. A.A.H., S.T., D.M., and S.S.N. designed the experiments. A.A.H., S.T., S.S.N., D.M., A.M.A., N.M., and B.L.Z. executed the experiments. S.S.N. developed the code and analyzed the data. A.A.H., H.T.S., M.E., A.R.D., S.S.N., S.T. wrote the paper. All authors edited, discussed, and approved the whole paper.

## Competing interests

The authors declare no competing interests.

## Additional information

**Peer review information** *Nature Communications* thanks Narain Karedla and other anonymous reviewer(s) to the peer review of this work. Peer review reports are available.

