## [Peer Review File · Nature Communications]

REVIEWER COMMENTS

Reviewer #1 (Remarks to the Author):

The authors have demonstrated a single-molecule coincidence immunoassay that improves the discrimination of specifically bound detection antibodies over non-specifically bound molecules that dominate backgrounds in many immunoassays. The approach seeks to address a fundamental challenge in improving the sensitivity of immunoassays—backgrounds from non-specific binding of detection antibodies—by using single molecule detection to only count binding events between detection and capture that are assumed to be mediated by the analyte in the immunocomplex. The paper is well written, the methods are clearly described, the experiments are well designed and controlled, and the data support the conclusions made by the authors. The demonstration of co-localization in an immunoassay is an impressive technical achievement. The primary hypothesis is demonstrated, i.e., that by specifically detecting co-localized molecules and normalizing them to the total number of capture antibodies the assay backgrounds can be lowered and precision improved. As a result, I recommend that this innovation warrants publication in Nature Communication pending some additional information (below).

While an impressive technical demonstration, the method does not improve the sensitivity of immunoassays, and is much less sensitive than other single-molecule immunoassays. The LODs reported are comparable to conventional immunoassays (pM) and do not approach those of other single-molecule immunoassays that have been reported (fM and below), such as ref. 11. The authors allude to this fact in the first sentence of the last paragraph of the Conclusions. They should be more explicit and quantitative about these comparisons to enable the reader to place assay performance in context. The authors do indicate that greater sensitivity might be achieved by imaging larger areas, but I think they also need to discuss the trade off with binding kinetics that this approach seems to have. To achieve ultra-sensitive immunoassays, high concentrations of capture antibodies (nM) need to be used over relatively small capture areas, usually by densely packing capture antibodies on surfaces. That approach does not seem feasible for the reported detection approach, as the capture antibodies must be spatially separated to resolve individual complexes and to have space between capture molecules to avoid non-specific bridging of detection antibodies across multiple capture antibodies. As such, the density of capture antibodies that can be used to kinetically drive binding is at odds with the detection method, so raises the questions whether sub-picomolar LODs can be achieved. This concern is highlighted by the need for overnight incubations with sample to achieve just picomolar sensitivities. It would be helpful for the authors to comment on this point, and provide quantitative information on the concentration of capture antibodies, the surface areas involved, and the implications to the kinetics of binding.

The authors should indicate the total imaging time per sample in the current set up and the envisioned larger area needed to improve sensitivity. They mention that each region of interest requires 128 images of the microscope's field of view to be acquired that only covers 0.5% of the capture area, but they do

not provide an imaging time. Imaging time is also important to present as the bound molecules will dissociate over the course of a measurement depending on off-rates. The authors should explain the impact of dissociation on these measurements.

The authors should also comment on the absolute CVs and not just the relative improvements of the different analysis methods. While the CVs were improved by colocalization and normalization, the absolute values (20% for serum (Fig. 3D)) are still high compared to conventional ELISA (<10%).

The plots in Figure 3B should include the experimental data with error bars not just the fits, as are plotted on Figure S7 to give the reader a sense of the variability in the measurement.

The dark spots that correspond to colocalized molecules are hard to see in Figure 2A. A zoomed in example would be very helpful.

Reviewer #2 (Remarks to the Author):

In this manuscript, the authors apply dual colour single-molecule colocalization using TIRF microscopy to enzyme-linked immunosorbent assays (ELISA) with the aim to improve its sensitivity and specificity. The authors present a clear motivation for the work they target in the manuscript, and in fact, discriminating true signal from non-specific background is one of the major limitations of ELISA deterring its ability to detect protein biomarkers and the binding constants of antibodies in a robust and reproducible manner. The success of a generalized method to improve on these aspects, such as the one presented in this work, will have a significant impact on both, fundamental research and diagnostic applications. The authors conduct a series of experiments using the same antibodies and biomarker in various conditions to explore the performance of Single-Molecule Colocalization Assay (SiMCA) in terms of eliminating the non-specific binding of detection antibodies, substrate to substrate variability, and determining the equilibrium dissociation constant of the biomarker for the chosen collection antibody. Whereas the ability to eliminate false positive is clear, the work presented here raises a few questions regarding the quantification of the results and the general broad applicability of the method as claimed in the introduction. It would be great if the authors could address these questions for the completeness of the message that they would like to deliver to the relevant communities:

- 1) The authors present results for a single biomarker with a correspondingly chosen collection antibody / detection antibody pair. For a general applicability of the method, can the authors experiment on a few biomarker antibody systems testing the suitability and limitations of the method for multivalent antigens/antibodies?

2) The limitations of using the 'sandwich' format where a minimum of two binding reactions are involved for a successful detection count are still present. Now, with the requirement of labelling both antibodies means that the labelling yield of each needs to be considered. The authors mention 90% labelling efficiency which means now the counts are underestimated by at least 20% if the binding affinities of both antibodies is significantly high. Can the results presented here, for at least one system be compared with any other benchmark technique such as SPR? Can the authors discuss this aspect in their article?

3) While performing single-molecule localization experiments, the concentration limits of the anchoring antibody might be necessary to be mentioned. Are these upper limits for the allowed concentration reasonable within the context of a typical ELISA experiment? Next, is there a minimum binding affinity between the protein/antibody to detect at all? Can the authors highlight any weak antibody/target biomarkers that cannot be used with this approach?

4) The authors correct for the chromatic aberrations using efficient image registration algorithms, but chose a Euclidean distance as large as 1.5 pixels that corresponds to roughly 200 nm. In most literature, fluorescent proteins or antibodies are considered to be colocalized when present within a distance range of 30 to 40 nm. See for example: Annibale et al., *Optical Nanoscopy* 1.1 (2012): 1-13, or perform statistical analysis based on image cross correlation methods or Ripley's k-function to refine a criterion (Thibault, et al., *Cytometry Part A* 87.6 (2015): 568-579). Can the authors comment on this? How would the results change if a tighter colocalization criteria that are used in literature applied here?

5) Can the authors comment on the relevance of the Bmax values reported if the concentration of the capturing antibody on the substrate is not known?

6) Can the authors help understand the main relevance of lower LOD values from a two-colour localization experiment? I understand that in general a TIRF illumination improves contrast and SNR similarly for both colour channels. If LOD of a single colour experiment is high because it includes false positives as well and removing these values lowers the LOD values, what is the main message that we gain from this parameter?

7) In summary, to throw some light into the field, can the authors comment on the applicability of their new approach in comparison to various existing single-molecule imaging based methods for biomarker detection such as SPR, interferometric scattering mass spectrometry (iSCAMS) etc.?

Reviewer #3 (Remarks to the Author):

The authors describe a clever method to overcome multiple problems with existing ELISA assays that stem from non-specific binding. The authors tackle this problem by tagging distinct optical probes to both the capture and the detection antibodies. That way, they can discriminate a true signal only if the two fluorophores of different colors coincide.

This way, they automatically rule out any signal coming from the detection antibody if the signal is "alone" and not combined with that of the capture antibody. The authors use rather sophisticated optical equipment to be able to resolve coincidence of fluorophores at the single molecule level, which is at the heart of the detection method.

The authors characterize their system experimentally and demonstrate that their main claim of the new colocalized assay lowering the variation between experiments is well founded.

I believe the manuscript can be strengthened by answering the following questions/addressing the concerns:

1. The 3x improvement in LOD is rather modest. The authors do state that LOD is not their main goal, but a discussion of why a higher improvement wasn't achieved would be helpful. Indeed, Figures 3B and C show that in both buffer and serum, the new assay is actually increasing the K_d , which normally should have an adverse effect on LOD. A discussion of LOD improving despite some increase in K_d would be helpful.

2. The authors demonstrate superior robustness (i.e. lower variation) in their assays as opposed to regular, single color assays. This reviewer would love to see a discussion of how significant the reduction in that variation is in the context of non-analytical variations that might affect the assay. For example, the authors use TNF α as an exemplary analyte to detect (and motivate it with patients that might undergo septic shock). How does the TNF α levels vary from patient to patient that exhibit similar characteristics? And if that variation is inherently high then, is it necessary to bring down the variability of the assay to level below that of conventional ELISA (especially when the new assay involves more sophisticated instrumentation?)

3. The paper can benefit from a more in depth analysis of the assays limitations. The authors do touch this at the very end of the paper but in my opinion it deserves its own section in the main body of the paper.

All in all, very nice approach to solve an important problem. The manuscript can be more ready for publication if the above points are addressed.

Reviewer 1: The reviewer recommended publication once we had addressed several important points and suggested some additional control experiments prior. We greatly appreciate the reviewer's thoughtful comments and have addressed these as detailed below:

1) The reviewer noted that we “**should be more explicit and quantitative**” in comparing the LODs achieved with SiMCA versus conventional and single-molecule immunoassays: “**While an impressive technical demonstration, the method does not improve the sensitivity of immunoassays, and is much less sensitive than other single-molecule immunoassays. The LODs reported are comparable to conventional immunoassays (pM) and do not approach those of other single-molecule immunoassays that have been reported (fM and below), such as ref. 11. It would be helpful for the authors to comment on this point.**”

We agree that our assay's sensitivity does not approach other single-molecule immunoassays such as the SiMoAs method described in Ref. 11. We have extended the Discussion/SI to explain these limitations of our assay as follows:

“Our focus in this work was to understand and reduce general sources of error in immunoassays, rather than to demonstrate sensitivity that outperforms existing molecular detection assays. We note that in theory the sensitivity of SiMCA is limited primarily by the number of dAbs and cAbs counted, as well as the accuracy with which colocalization is determined. The former quantity can be addressed simply by scanning larger FOVs on the coverslip; in the present study, we examined only 0.5% of the flow cell surface. In our current set-up, the EMCCD camera's field of view was adjusted to split the channels to show parallel images for single-molecule Förster resonance energy transfer (FRET) experiments (described below). Scanning larger areas is also possible but may impact the assay's performance by increasing imaging time, such that the effects of dissociation/off-rate will become more meaningful. In future studies, we will explore a post-crosslinking approach, which will enable us to scan coverslips for tens of minutes and thereby achieve better sensitivity. Increasing colocalization accuracy helps to eliminate false positives, while also allowing higher cAb densities (**SI Fig. S11**). In its current form (**SI Fig. S11, left**), the assay's primary limitation on sensitivity is its low [cAb] (~2 pM), resulting in low capture efficiency of target molecules²⁵. This low [cAb] is required in SiMCA's current format due to the coating density limitations imposed by diffraction-based identification and localization of cAbs. Future approaches that enable higher effective [cAb] levels (**SI Fig. S11, right**), and thus higher target capture efficiencies, could be achieved by strategies including increased surface area-to-chamber volume ratios and super-resolution imaging techniques that achieve sub-diffraction-limited molecular localization (*e.g.* STORM/STED²⁶, DNA-PAINT²⁷, FRET²⁸, etc.).

As single-molecule fluorescence colocalization can be determined with single-Angstrom precision, this suggests that the ultimate limit on colocalization in our assay is the size of the antibodies used (~10 nm). We note as well that FRET provides an alternate, stringent test of fluorophore colocalization. Indeed, we observed FRET between colocalized dAbs and cAbs (**SI Fig. S12**), and we are now exploring the use of FRET to increase the sensitivity and specificity of SiMCA. Finally, we would note that SiMCA, like other immunoassays approaching single-molecule detection, is limited by molecular shot noise, where the theoretical sensitivity is statistically dictated by unavoidable Poisson error²⁹.” (**See Discussion, Page 14**)

Figure S11: Predicted equilibrium binding (capture efficiency) with an estimated effective [cAb] of 2 pM (**left**). Increasing cAb density improves target capture efficiency by more than 100-fold (**right**). Note that these plots assume that all captured TNF- α gets labeled by a dAb, where [dAb] > 50 nM and [TNF- α] \geq 10 pM, such that we are not limited by depletion effects. This is why the plots were generated only by using the first equilibrium reaction (see Ref. 25).

We have also added a detailed discussion explicitly comparing SiMCA to SiMOAs, iSCAT, iSCAMS, and SPR techniques (**See Discussion, Page 15**). We discuss the applicability of our approach in comparison to existing single-molecule imaging-based methods for biomarker detection, and explicitly compared performance with a focus on robustness against non-specific background. We also provide an in-depth analysis of our assay's limitations. Unfortunately, we cannot quantitatively benchmark SiMCA against ELISA, SPR, and other assays, since SiMCA is an intrinsically different assay.

2) The reviewer requested quantitative information on the concentration of capture antibodies, the surface areas involved, and the implications to the kinetics of binding. The reviewer also asked about the total imaging time per sample, the envisioned larger area needed to improve sensitivity, and the impact of dissociation on these measurements: “Provide quantitative information on the concentration of capture antibodies, the surface areas involved, and the implications to the kinetics of binding. The authors should also indicate the total imaging time per sample in the current set up and the envisioned larger area needed to improve sensitivity. They mention that each region of interest requires 128 images of the microscope’s field of view to be acquired that only covers 0.5% of the capture area, but they do not provide an imaging time. Imaging time is also important to present as the bound molecules will dissociate over the course of a measurement depending on off-rates. The authors should explain the impact of dissociation on these measurements.”

We have added quantitative information to address the reviewer’s questions, including the following addition to the Methods section:

“We used a custom-made polycarbonate chamber with dimensions of 13 mm x 4 mm x 150 μm , which matches the size of the cAb-functionalized area. The concentration of cAbs on the surface was estimated to be around 2 pM. We collected assay data by rastering a 400- μm x 400- μm area of the pegylated region of the coverslip at 5 μm intervals, producing 64 images per coverslip per channel. Two sets of 64 images were collected for each TNF- α concentration. The collection of these images, which each included green (200 ms) and red (200ms) channels (200 ms) plus a blank (for oil equilibration, 2.5 s), took \sim 3 min in total, and provided sufficient precision for all samples tested over the 0.01–10 nM TNF- α concentration range studied (**SI Figure S13** shows the calculated CVs of dAb and cAb counts after sampling 3–100 FOVS for eight coverslips (bootstrapped). We can see that a plateau is reached far well before the 64 FOV mark.

We limited the number of FOVs (64 FOVs, 0.5% of the flow cell surface) scanned to minimize imaging time (\sim 3 min)—and thereby minimize the effects of dissociation on the sensitivity of our assay. We are aware that scanning larger FOVs on the coverslip would improve assay sensitivity. One solution would be to switch to a set-up with a larger FOV; this would be challenging with our current set-up, in which the EMCCD camera FOV has been adjusted to split the channels to show parallel images for single-molecule FRET experiments. Scanning larger areas is also possible, but may impact assay performance by increasing imaging time to an extent that dissociation/off-rate becomes a meaningful factor.

Using SPR, flow cytometry, and previously reported data on the $K_d/K_{\text{on}}/K_{\text{off}}$ of the antibodies^{37,38}(mAb1 and mAb11) utilized in this study, the measured K_d values for mAb11 and mAb1 are in the range of \sim 1.0—2.0 nM (**SI Fig. S14**) and the reported average K_{on} for an antibody is \sim 10⁵ M⁻¹s⁻¹. With such high affinities, we expect minimal dissociation ($K_d = K_{\text{off}}/K_{\text{on}}$) within the reported imaging time (\sim 3 min). However, as mentioned in the Discussion (**See Discussion, Page 14**), we are exploring the use of a post-crosslinking approach that will enable us to scan coverslips for tens of minutes and thereby achieve better sensitivity.”

Figure S13: Calculated CVs of dAb and cAb counts after sampling 3–100 FOVs for eight coverslips (bootstrapped). A plateau is reached well before the 64 FOV mark.

Figure S14: Estimated K_d of mAb1 using SPR and flow cytometry techniques. See Reference 37 for more information on mAb11 K_d .

3) The reviewer asked us to comment on absolute CVs, noting that the absolute values (20% for serum (Fig. 3D)) are still high compared to conventional ELISA (<10%): “The authors should also comment on the absolute CVs and not just the relative improvements of the different analysis methods. While the CVs were improved by colocalization and normalization, the absolute values (20% for serum (Fig. 3D)) are still high compared to conventional ELISA (<10%).”

We would like to point out that the percentages shown in **Figure 3D** are the Mean-Absolute-Percentage-Log-Error (MAPLE) for quantification from bootstrapped sets, and not absolute CV values. We have updated the caption to clarify this and thank the reviewer for highlighting this potential point of confusion. Conventional reports of error in ELISA do not report the MAPLE score. To address the reviewer’s question, we calculated absolute CVs for single-color and colocalized normalized counts in both serum and buffer in the table below (outliers removed). The slightly higher CV in normalized colocalized counts relative to the <10% cited by the reviewer for conventional ELISA could be a function of our basic image segmentation techniques and other aspects of the assay such as unoptimized dAb conditions, hand pipetting of reagents and manual assembly of coverslips, and the use of glass coverslips with differences in surface chemistry preparation. These factors could collectively lead to variance due to small dilution errors or small changes in how the samples get added and washed from the chambers. However, we would like to stress that relative improvement of CVs or stability of CVs across methods and sample matrices is more important to understanding and improving overall assay functionality than absolute CVs.

	Buffer	Serum
1-color	37%	16%
Colocalized	13%	13%

4) The reviewer requested inclusion of experimental data with error bars in Fig. 3B, and a zoomed-in example from Figure 2A to better show dark spots corresponding to colocalized molecules: “The plots in Figure 3B should include the experimental data with error bars not

just the fits, as are plotted on Figure S7 to give the reader a sense of the variability in the measurement.”

We thank the reviewer for their comment. We have modified **Figure 3B** to include the error bars as suggested (see below).

We have also provided in **SI Figure S4** an enlargement of **Figure 2A** that more clearly displays the better resolution of non-specifically bound spots and their minimal degree of colocalization in $\text{TNF-}\alpha$ -free control samples.

Figure S4: Enlarged single-color fluorescence images of dAb only (top), two-color images of cAb and dAb (middle), and log-scale inverted composite images of two-color detection (bottom) in the absence of TNF- α and with 50 nM (left) or 500 nM (right) dAb. Dark spots in the bottom panels represent colocalized signal from the two fluorophores.

Reviewer 2: The reviewer felt that our method could have a significant impact on both fundamental research and diagnostic applications, but also asked that we address several questions to improve the completeness of the work. We greatly appreciate the reviewer’s thoughtful comments and recommendations, and have addressed them as detailed below:

1) The reviewer asked if we could test other biomarker antibody systems to assess the generalizability of our method: “The authors present results for a single biomarker with a correspondingly chosen collection antibody / detection antibody pair. For a general applicability of the method, can the authors experiment on a few biomarker antibody systems testing the suitability and limitations of the method for multivalent antigens/antibodies?”

We thank the reviewer for this suggestion. To prove the general applicability of the SiMCA method, we functionalized another cAb/dAb pair targeting the MCP-1 protein using the same commercially-available fluorophore and biotin kits (See **Methods** and **SI Figure S8**). As with TNF- α , we again demonstrated the capability of SiMCA to eliminate the confounding effects of background produced by non-specific binding of dAbs, and greatly improved the reproducibility of our measurements, with a pM detection limit.

Figure S8: Generalizability of SiMCA to a second protein analyte, monocyte chemoattractant protein-1 (MCP-1). Just like TNF- α , we passivated a glass coverslip with a mixture of PEG and PEG-biotin, and then treated with neutravidin and biotinylated, Alexa-546-tagged capture antibodies (cAbs). The surface is then incubated with a solution of the MCP-1 target biomolecule and Alexa-647-labeled detection antibody (dAb) (**See Methods**). A) Single-color fluorescence images of dAb only (top), two-color images of cAb and dAb (middle), and log-scale inverted composite images of two-color detection (bottom) in the absence of MCP-1 with 50 nM (left) or 800 nM (right) dAb. Dark spots in the bottom panels represent colocalized signal from the two fluorophores. B) Distributions of absolute single-color and colocalized counts across 128 fields of view (FOVs). Dashed lines demarcate quartiles of the distribution. C) Absolute number of dAbs per fields of view. D) Normalized, colocalized counts across different coverslips and MCP-1 concentrations. Each violin represents 128 FOVs per coverslip (64 per channel).

2) The reviewer asked about the impact of antibody labeling yield on detection performance, and if we could compare our results with another benchmark technique such as SPR: “The limitations of using the ‘sandwich’ format where a minimum of two binding reactions are involved for a successful detection count are still present. Now, with the requirement of labelling both antibodies means that the labelling yield of each needs to be considered. The authors mention 90% labelling efficiency which means now the counts are underestimated by at least 20% if the binding affinities of both antibodies is significantly high. Can the results presented here, for at least one system be compared with any other benchmark technique such as SPR? Can the authors discuss this aspect in their article?”

We apologize for the confusion, as we overlooked important quantitative information regarding the degree of labelling (DOL) for each antibody in our original manuscript, and provided an inaccurate average DOL instead (*i.e.*, 90% which was the average probability of having **more** than 1 dye for mAb1/mAb11 antibodies). We have now corrected this number and detailed the DOL of both the dAbs and cAbs, which were purchased from Sigma-Aldrich and have well-established labeling protocols. We have also added a table computing the Poisson distribution (calculated by $P(x, \text{DOL}) = (e^{-\text{DOL}}) (\text{DOL})^x / x!$) of dye labels per antibody based on that DOL to the Methods section (**Table 3**). The probability of having no dyes for each antibody is close to zero, and the detection of even single dye labels using single-molecule fluorescence is routine.

Table 3: Poisson distribution of the number of dye labels per antibody.

	Alexa 546 bio-mAb1	Alexa 647- mAb11	Alexa 647 - 10F7	Alexa 546 Bio - 5F3D7
Degree of Labelling	4	3.9	4.4	4.7
Dye per Ab				
0	1.83%	2.02%	1.23%	0.91%
1	7.33%	7.89%	5.40%	4.27%
2	14.65%	15.39%	11.88%	10.05%
3	19.54%	20.01%	17.43%	15.74%
4	19.54%	19.51%	19.17%	18.49%
5	15.63%	15.22%	16.87%	17.38%
6	10.42%	9.89%	12.37%	13.62%

7	5.95%	5.51%	7.78%	9.14%
8	2.98%	2.69%	4.28%	5.37%
9	1.32%	1.16%	2.09%	2.81%
10	0.53%	0.45%	0.92%	1.32%

As for the second part of the reviewer's question, we have added the following comparison between our SiMCA approach and other state-of-the-art techniques like SPR to the Discussion section:

“SPR has been widely used for the measurement of biomolecular interaction kinetics in real-time. In SPR, we can detect changes in the reflected light when an analyte binds to (or unbinds from) the sensor surface, making this technique label-free and direct. This method utilizes total internal reflection to achieve high sensitivity by exciting surface plasmons at a critical angle within a penetration depth of around 100 nm. Immobilization of the analyte-binding substrate in SPR is achieved by adsorption onto gold surfaces, in contrast to SiMCA, which uses PEG passivation and biotin-streptavidin to specifically immobilize cAbs. Although both techniques differ with their immobilization strategies, SPR and SiMCA offer similar sensitivities for detection. Furthermore, because SPR is a label-free method, it cannot confidently distinguish between specific binding of the analyte versus other biomolecules from serum, blood, or other complex media. Finally, SiMCA is a two-color fluorescence-based detection assay while SPR is a single-binder assay, and thus offers less specificity and higher background noise. Moreover, reliance on fluorescence for identification makes SiMCA apt for detecting analytes directly from serum and blood by filtering out non-specific binding arising from various biomolecules present in these complex media through two-color colocalization.” (See Pages 15-16, Discussion)

3) The reviewer asked about the concentration limits of the cAb, and whether these are reasonable for a typical ELISA experiment. They also asked if there is a minimum binding affinity needed for detection: “While performing single-molecule localization experiments, the concentration limits of the anchoring antibody might be necessary to be mentioned. Are these upper limits for the allowed concentration reasonable within the context of a typical ELISA experiment? Next, is there a minimum binding affinity between the protein/antibody to detect at all? Can the authors highlight any weak antibody/target biomarkers that cannot be used with this approach?”

Most antibodies bind strongly to their target antigen, with K_D values in the low micromolar to nanomolar range, although high-affinity antibodies can exhibit K_D values in the low nanomolar or even picomolar range. Per the reviewer's suggestion, we have added to the Discussion the range of affinities for sandwich antibodies that will work with SiMCA. Most antibodies with at least sub-micromolar K_D values can be used with our approach. However, lower affinity antibodies with K_D in the micromolar-millimolar range create legitimate concern that the low cAb surface coverage might impede SiMCA performance. This could be an issue for detecting certain ligands such as some small molecules and peptides. In order to generalize our assay to low-affinity antibodies, we will be looking into various potential solutions including: 1) increasing cAb density, which will

result in higher capture efficiency (avidity) and a lower effective off-rate, 2) decreasing imaging time, and 3) improving assay kinetics through the use of crowding agents or cross-linking of antibody-target pairs. We have added this information to the Methods and Discussion sections of the revised manuscript (See **Discussion, Page 15**).

4) The reviewer asked about the 1.5 pixel/200 nm distance employed in our image registration algorithms, and noted that prior work has used a threshold of 30 to 40 nm. They also asked how the results might change if tighter colocalization criteria were applied: “The authors correct for the chromatic aberrations using efficient image registration algorithms, but chose a Euclidean distance as large as 1.5 pixels that corresponds to roughly 200 nm. In most literature, fluorescent proteins or antibodies are considered to be colocalized when present within a distance range of 30 to 40 nm. See for example: Annibale et al., *Optical Nanoscopy* 1.1 (2012): 1-13, or perform statistical analysis based on image cross correlation methods or Ripley’s k-function to refine a criterion (Thibault, et al., *Cytometry Part A* 87.6 (2015): 568-579). Can the authors comment on this? How would the results change if a tighter colocalization criteria that are used in literature applied here?”

We thank the reviewer for this thoughtful comment, and have edited the Methods section to include the following text:

“By implementing tighter colocalization criteria of 10–40 nm, either statistically and/or with advanced imaging, we would expect drastically improved sensitivity and a decrease in the number of false-positive events. This is also influenced by the amount of cAb on the coverslip surface. Higher cAb density improves assay sensitivity by providing more binding sites, but will also confound the discrimination of fluorescent spots and introduce errors in single-molecule counting in a diffraction-limited set-up. To determine the balance of optimal spatial resolution and capture density needed to maximize the sensitivity of our assay, we performed simulations for different colocalization cutoff distances (10–300 nm) to estimate the number of false colocalization events as the number of non-specific binding events increases (**SI Fig S17**). Shorter distances would reduce the number of false colocalizations, but also requires higher spatial resolution. For example, with our current colocalization criteria (~200 nm), we would expect ~4.5 of every 100 non-specific binding events to be counted as a binding event. With a distance of 100 nm, we would estimate this number to be ~1— theoretically reaching 0 at a distance of 10 nm. We also confirmed that increasing the number of cAbs on the surface would increase the number of false colocalizations, but could also lower the assay LOD. The smallest simulated cutoff distance (10 nm) could be achieved through FRET-based detection or with super-resolution techniques and more sophisticated software and imaging techniques.”

This latter work is ongoing, however, and beyond the scope of the present manuscript.

Figure S17: Simulations of different colocalization cutoff distances (10–300 nm) to estimate the number of false colocalization events as the number of non-specific binding events increases. Shorter distances (10nm) correspond to a stricter colocalization cutoff which would reduce the number of false colocalizations but requires higher spatial resolution (Top panels). We showed that by increasing the number of capture antibodies on the surface, we would increase the number of false colocalizations but can also lower the limit of detection of the assay (Bottom panels).

5) The reviewer asked about the relevance of the Bmax values reported if the concentration of the capturing antibody on the substrate is not known: “Can the authors comment on the relevance of the Bmax values reported if the concentration of the capturing antibody on the substrate is not known?”

The estimation of Bmax does not depend on knowing the concentration of the cAb. This parameter is derived from the binding curve data, which was calculated as described in Methods. Bmax represents the maximum signal possible—or in the case of spot counting, the total number of binding regions available on the coverslip surface. In the case of unnormalized single-color counts, this number encompasses all non-specific binding regions as well as cAbs. For colocalized counts, which are also unnormalized since we are assuming we do not know the cAb concentration, the

fitted Bmax value would be an estimate of the total number of cAbs available per FOV. Normalized colocalized counts would, by definition, require knowing the number of cAbs. Thus, in this case, Bmax represents the maximum proportion of binding sites that dAbs would bind to out of all available cAbs.

6) The reviewer asked if LOD of a single colour experiment is high because it includes false positives as well and removing these values lowers the LOD values, what is the main message that we gain from this parameter?: “Can the authors help understand the main relevance of lower LOD values from a two-colour localization experiment? I understand that in general a TIRF illumination improves contrast and SNR similarly for both colour channels. If LOD of a single colour experiment is high because it includes false positives as well and removing these values lowers the LOD values, what is the main message that we gain from this parameter?”

We surpass the fundamental limitations of SNR in TIRF by combining information from two colors and ensuring that spots showing up in both channels are colocalized in order to be counted as a real binding event. By discarding false-positive dAb signals that are not colocalized with a cAb signal, we can greatly decrease the background noise, which improves the SNR. In this fashion, we can lower the LOD and thereby increase the sensitivity of the assay for detecting analytes at lower concentrations.

7) The reviewer asked that we comment on the applicability of our approach in comparison to existing single-molecule imaging-based methods for biomarker detection: “In summary, to throw some light into the field, can the authors comment on the applicability of their new approach in comparison to various existing single-molecule imaging based methods for biomarker detection such as SPR, interferometric scattering mass spectrometry (iSCAMS) etc.?”

Based on the reviewer’s suggestion, we have added the following passages to our Discussion:

“Despite these limitations, our SiMCA approach has multiple advantages in comparison to other single-molecule imaging-based methods such as SiMoAs, iSCAT, iSCAMS and other state-of-the-art techniques like SPR—most notably, its robustness against non-specific binding.³¹⁻³⁴ SiMoAs immunoassays capture microscopic beads decorated with specific antibodies and then label the immunocomplexes with an enzymatic reporter capable of generating a fluorescent product. After isolating the beads in 50 fL reaction chambers designed to hold only a single bead, fluorescence imaging is used to detect single protein molecules. This approach can detect as few as ~10–20 enzyme-labeled complexes in a 100 µl sample and allows detection of clinically relevant proteins in serum at femtomolar concentrations and less, which is much lower than conventional ELISA. However, unlike SiMCA, this assay does not provide the ability to isolate and interrogate single molecules on individual beads, and thereby distinguish true antibody-antigen binding events from non-specifically bound complexes. SiMCA also offers a simpler assay approach by comparison.

Methods such as iSCAT, iSCAMS and SPR offer the powerful advantage of label-free imaging. iSCAT and iSCAMS are inexpensive alternatives that deliver real-time imaging of single

unlabeled biomolecules in their natural environment. When illuminated with coherent light, biomolecules in solution both scatter and reflect light, with scattered light being the most pronounced. In iSCAT and iSCAMS, the scattered and reflected light interfere at the detector, giving rise to contrast and allowing single particles to be imaged. Interferometric signal contrast relies on the intensity balance between the molecule-induced scattered light and the reference laser beam, apart from the scattering cross-section of the molecule itself. However, high illumination intensity is needed for the molecule-scattered photons to overcome background noise. Importantly, the lack of specificity to differentiate nano-objects beyond the intensity of the signal and the characteristic dynamic behavior of the object under study poses a considerable experimental challenge when dealing with complex, multicomponent systems.

SPR has been widely used for the measurement of biomolecular interaction kinetics in real-time. In SPR, we can detect changes in the reflected light when an analyte binds to (or unbinds from) the sensor surface, making this technique label-free and direct. This method utilizes total internal reflection to achieve high sensitivity by exciting surface plasmons at a critical angle within a penetration depth of around 100 nm. Immobilization of the analyte-binding substrate in SPR is achieved by adsorption onto gold surfaces, in contrast to SiMCA, which uses PEG passivation and biotin-streptavidin to specifically immobilize cAbs. Although both techniques differ with their immobilization strategies, SPR and SiMCA offer similar sensitivities for detection. Furthermore, because SPR is a label-free method, it cannot confidently distinguish between specific binding of the analyte versus other biomolecules from serum, blood, or other complex media. Finally, SiMCA is a two-color fluorescence-based detection assay while SPR is a single-binder assay, and thus offers less specificity and higher background noise. Moreover, reliance on fluorescence for identification makes SiMCA apt for detecting analytes directly from serum and blood by filtering out non-specific binding arising from various biomolecules present in these complex media through two-color colocalization.” (See Discussion, Page 15-16)

Reviewer 3: The reviewer felt that our approach can overcome multiple problems with existing ELISA assays, but raised several important points as well. We greatly appreciate the reviewer's thoughtful comments and recommendations, and have addressed them as detailed below:

1) The reviewer asked why we only achieved a modest three-fold improvement in LOD, and why LOD was improved despite seeing an increase in K_D : “The 3x improvement in LOD is rather modest. The authors do state that LOD is not their main goal, but a discussion of why a higher improvement wasn't achieved would be helpful. Indeed, Figures 3B and C show that in both buffer and serum, the new assay is actually increasing the K_D , which normally should have an adverse effect on LOD. A discussion of LOD improving despite some increase in K_D would be helpful.”

We thank the reviewer for this question, and have added the following explanation to the Discussion section:

“Our focus in this work was to understand and reduce general sources of error in immunoassays, rather than to demonstrate sensitivity that outperforms existing molecular detection assays. We note that in theory the sensitivity of SiMCA is limited primarily by the number of dAbs and cAbs counted, as well as the accuracy with which colocalization is determined. The former quantity can be addressed simply by scanning larger FOVs on the coverslip; in the present study, we examined only 0.5% of the flow cell surface. In our current set-up, the EMCCD camera's field of view was adjusted to split the channels to show parallel images for single-molecule Förster resonance energy transfer (FRET) experiments (described below). Scanning larger areas is also possible but may impact the assay's performance by increasing imaging time, such that the effects of dissociation/off-rate will become more meaningful. In future studies, we will explore a post-crosslinking approach, which will enable us to scan coverslips for tens of minutes and thereby achieve better sensitivity. Increasing colocalization accuracy helps to eliminate false positives, while also allowing higher cAb densities (SI Fig. S11). In its current form (SI Fig. S11, left), the assay's primary limitation on sensitivity is its low [cAb] (~2 pM), resulting in low capture efficiency of target molecules²⁵. This low [cAb] is required in SiMCA's current format due to the coating density limitations imposed by diffraction-based identification and localization of cAbs. Future approaches that enable higher effective [cAb] levels (SI Fig. S11, right), and thus higher target capture efficiencies, could be achieved by strategies including increased surface area-to-chamber volume ratios and super-resolution imaging techniques that achieve sub-diffraction-limited molecular localization (*e.g.* STORM/STED²⁶, DNA-PAINT²⁷, FRET²⁸, etc.).

As single-molecule fluorescence colocalization can be determined with single-Angstrom precision, this suggests that the ultimate limit on colocalization in our assay is the size of the antibodies used (~10 nm). We note as well that FRET provides an alternate, stringent test of fluorophore colocalization. Indeed, we observed FRET between colocalized dAbs and cAbs (SI Fig. S12), and we are now exploring the use of FRET to increase the sensitivity and specificity of SiMCA. Finally, we would note that SiMCA, like other immunoassays approaching single-molecule detection, is limited by molecular shot noise, where the theoretical sensitivity is statistically dictated by unavoidable Poisson error²⁹.” (See Discussion, Page 14)

As for the second part of the reviewer's question, this assay is not accurately described at the physical level by a strict Langmuir isotherm. This is because each molecular detection event

involves two separate binding events, and so the law of mass action dictates the dose-response curve here. Thus, the K_D values obtained are not strictly affinity constants. One would also expect that the K_D values obtained from Langmuir fitting would be higher for the colocalized analysis method, as there will be fewer observed colocalized signals than dAb-only signals; in the Langmuir equation, this would indicate dAbs having a higher affinity for the surface than colocalized ternary complexes.

2) The reviewer asked how significant the reduction in variation achieved with our colocalization method is in the context of non-analytical variations that might affect the assay, citing the example of how TNF α levels vary across septic shock patients exhibiting similar characteristics: “The authors demonstrate superior robustness (i.e. lower variation) in their assays as opposed to regular, single color assays. This reviewer would love to see a discussion of how significant the reduction in that variation is in the context of non-analytical variations that might affect the assay. For example, the authors use TNF α as an exemplary analyze to detect (and motivate it with patients that might undergo septic shock). How does the TNF α levels vary from patient to patient that exhibit similar characteristics? And if that variation is inherently high then, is it necessary to bring down the variability of the assay to level below that of conventional ELISA (especially when the new assay involves more sophisticated instrumentation?)”

We thank the reviewer for this comment, as we are indeed very interested in applying the assay in clinical settings in the future. However, we would like to note that clinical diagnostics is not the focus of the present work, as we are focused primarily on developing capabilities rather than broader utility. Our study simply used TNF- α as a well-characterized model protein to demonstrate the feasibility and performance of our assay, and we have now added a second target (MCP-1, See **SI Figure S8**) to further demonstrate the generalizability of the approach.

3) The reviewer requested more in-depth analysis of the assay’s limitations: “The paper can benefit from a more in depth analysis of the assays limitations. The authors do touch this at the very end of the paper but in my opinion it deserves its own section in the main body of the paper.”

Per the reviewer’s suggestion, we have added several paragraphs to the Discussion section examining the assay’s limitations:

“As with any assay, SiMCA does suffer from some limitations. At present, SiMCA requires relatively expensive microscopy equipment that can achieve single-molecule sensitivity. Extending the benefits of SiMCA to lower-resource environments would require strategies to boost the fluorescence signal to levels that can be detected by smartphone cameras²³—for example, by using fluorescent nanoparticles that emit a substantially brighter signal, or fluorescence-enhancing materials²⁴ that maximize the output from individual fluorophores.

Our focus in this work was to understand and reduce general sources of error in immunoassays, rather than to demonstrate sensitivity that outperforms existing molecular detection assays. We note that in theory the sensitivity of SiMCA is limited primarily by the number of dAbs and cAbs counted, as well as the accuracy with which colocalization is determined. The former quantity can be addressed simply by scanning larger FOVs on the coverslip; in the present study, we examined

only 0.5% of the flow cell surface. In our current set-up, the EMCCD camera's field of view was adjusted to split the channels to show parallel images for single-molecule Förster resonance energy transfer (FRET) experiments (described below). Scanning larger areas is also possible but may impact the assay's performance by increasing imaging time, such that the effects of dissociation/off-rate will become more meaningful. In future studies, we will explore a post-crosslinking approach, which will enable us to scan coverslips for tens of minutes and thereby achieve better sensitivity. Increasing colocalization accuracy helps to eliminate false positives, while also allowing higher cAb densities (**SI Fig. S11**). In its current form (**SI Fig. S11, left**), the assay's primary limitation on sensitivity is its low [cAb] (~2 pM), resulting in low capture efficiency of target molecules²⁵. This low [cAb] is required in SiMCA's current format due to the coating density limitations imposed by diffraction-based identification and localization of cAbs. Future approaches that enable higher effective [cAb] levels (**SI Fig. S11, right**), and thus higher target capture efficiencies, could be achieved by strategies including increased surface area-to-chamber volume ratios and super-resolution imaging techniques that achieve sub-diffraction-limited molecular localization (*e.g.* STORM/STED²⁶, DNA-PAINT²⁷, FRET²⁸, etc.).

As single-molecule fluorescence colocalization can be determined with single-Angstrom precision, this suggests that the ultimate limit on colocalization in our assay is the size of the antibodies used (~10 nm). We note as well that FRET provides an alternate, stringent test of fluorophore colocalization. Indeed, we observed FRET between colocalized dAbs and cAbs (**SI Fig. S12**), and we are now exploring the use of FRET to increase the sensitivity and specificity of SiMCA. Finally, we would note that SiMCA, like other immunoassays approaching single-molecule detection, is limited by molecular shot noise, where the theoretical sensitivity is statistically dictated by unavoidable Poisson error²⁹.

Most antibodies with sub-micromolar K_D values should be compatible with SiMCA. However, with lower-affinity antibodies in the micromolar-millimolar range, there is a legitimate concern that the low cAb surface coverage might impede SiMCA performance. This could be an issue for detecting certain ligands, such as some small molecules and peptides. In order to generalize our assay to low-affinity antibodies, we will be looking into potential solutions including: 1) increasing cAb density, which will result in higher capture efficiency (avidity³⁰) and a lower effective off-rate, 2) decreasing imaging time, and 3) improving assay kinetics through the use of crowding agents or cross-linking of antibody-target pairs.” (**See Discussion, Page 14-15**)

REVIEWERS' COMMENTS

Reviewer #1 (Remarks to the Author):

The authors have satisfactorily addressed the critiques of the three reviewers in their revised manuscript and rebuttal letter. The manuscript is acceptable for publication.

Reviewer #2 (Remarks to the Author):

The authors have significantly enhanced the manuscript and the supporting information answering to all the comments and suggestions in the first revision round. The work will be of significant value in analytical biochemistry and diagnosis. The methodology presented is reproducible, flawless and will lead to technological advances in the relevant fields. I would therefore recommend for its publication in the journal.

Reviewer #3 (Remarks to the Author):

I believe the authors did a good job answering my questions, as well as modifying the paper accordingly. From my perspective, the manuscript is ready for publication.

REVIEWERS' COMMENTS

Reviewer #1 (Remarks to the Author):

The authors have satisfactorily addressed the critiques of the three reviewers in their revised manuscript and rebuttal letter. The manuscript is acceptable for publication.

Response: We thank the reviewer for their thoughtful and very constructive feedback.

Reviewer #2 (Remarks to the Author):

The authors have significantly enhanced the manuscript and the supporting information answering to all the comments and suggestions in the first revision round. The work will be of significant value in analytical biochemistry and diagnosis. The methodology presented is reproducible, flawless and will lead to technological advances in the relevant fields. I would therefore recommend for its publication in the journal.

Response: We thank the reviewer for their thoughtful and very constructive feedback.

Reviewer #3 (Remarks to the Author):

I believe the authors did a good job answering my questions, as well as modifying the paper accordingly. From my perspective, the manuscript is ready for publication.

Response: We thank the reviewer for their thoughtful and very constructive feedback.